# Object Affordance-Based Implicit Interaction for Wheelchair-Mounted Robotic Arm Using a Laser Pointer

**DOI:** 10.3390/s23094477

**Published:** 2023-05-04

**Authors:** Yaxin Liu, Yan Liu, Yufeng Yao, Ming Zhong

**Affiliations:** State Key Laboratory of Robotics and System, Harbin Institute of Technology, Harbin 150001, China; liuyaxin@hit.edu.cn (Y.L.); 21b908088@stu.hit.edu.cn (Y.L.);

**Keywords:** implicit interaction, WMRA, intention reasoning, conditional random field

## Abstract

With the growth of the world’s population, limited healthcare resources cannot provide adequate nursing services for all people in need. The wheelchair-mounted robotic arm (WMRA) with interactive technology could help to improve users’ self-care ability and relieve nursing stress. However, the users struggle to control the WMRA due to complex operations. To use the WMRA with less burden, this paper proposes an object affordance-based implicit interaction technology using a laser pointer. Firstly, a laser semantic identification algorithm combined with the YOLOv4 and the support vector machine (SVM) is designed to identify laser semantics. Then, an implicit action intention reasoning algorithm, based on the concept of object affordance, is explored to infer users’ intentions and learn their preferences. For the purpose of performing the actions about task intention in the scene, the dynamic movement primitives (DMP) and the finite state mechanism (FSM) are respectively used to generalize the trajectories of actions and reorder the sequence of actions in the template library. In the end, we verified the feasibility of the proposed technology on a WMRA platform. Compared with the previous method, the proposed technology can output the desired intention faster and significantly reduce the user’s limb involvement time (about 85%) in operating the WMRA under the same task.

## 1. Introduction

Nowadays, the world’s population is in the stage of growing [1]. The increasing number of people challenges the social care services system, and most care workers are facing unprecedented pressure [2]. The WMRA can help people complete some household tasks independently [3], which will improve the quality of life and reduce caregiving stress. However, the traditional joystick remote control mode on the WMRA requires frequent limb movements of the user, which brings physical and psychological burdens. The WMRA should be easy to operate and have a sense of control [4,5]. Therefore, helping users interact with the WMRA with less limb movements and conveying their intentions with the least operation is the current research focus impacting the performance and user acceptance of the WMRA.

To facilitate the operation of robots for users with different levels of physical ability, many human–robot interactions (HRI) interfaces utilizing residual limb abilities have been studied, such as using chin [6], shoulder [7], gesture [8], and eye movement [9]. In these studies, the interaction operations mapped the residual limb movement to robot instructions, such as forward, back, left, right, rotation, or other cartesian motions for the robotic arm, and some preset simple household tasks, which could help users perform some structured tasks, but still needing frequent limb movement. In addition, using biological signals remote control robots to perform the tasks would substantially reduce limb movements and physical burden [10,11,12,13]. However, the time-consuming signal recognition, complex operation, and high cost limit their development.

In recent years, researchers have tried to use the concept of “shared attention [14]” to reduce the complexity of manipulating robots. The concept refers to “what you see is what you want” so that robots can grasp or place the target objects for a user by capturing the focus of the user’s attention. For instance, screen tapping [15], eye gaze [16], laser pointer [17,18], and electroencephalogram (EEG) recognition [19] can share attention with robots. However, compared with laser pointer, gaze, EEG, and a screen would attract too much attention from users, which will make them ignore the surrounding environment and lead to safety risks.

Users could convey simple intentions to the robot through share attention. However, people’s intentions are complex and robots’ tasks are unstructured, which requires a robot to make deeper inferences about intention from a simple interaction. The concept of object affordance refers to actions that match the physical properties of an object [20,21], which contribute to reasoning the intention of deeper tasks from the shared attention of users. Therefore, we propose an implicit intention interaction technology using a laser pointer based on the idea of object affordance. A user could point to the objects in the scene using a laser pointer, and the WMRA would reason the action intentions through the selected objects and execute the corresponding action content.

The proposed technology is composed of two major parts. One is the laser semantics identification model based on the SVM and YOLOv4 fusion algorithm, which mainly identifies the corresponding semantics by detecting the flicker frequency of the laser spot. The other is the household task intentions reasoning model based on the conditional random field (CRF) and Q-learning algorithm. We formalize the concept of object affordance based on the CRF algorithm and reason the user’s intention through the object information in the scene. In addition, through the Q-learning algorithm, the proposed reasoning model can learn user preferences in the long-term historical intention prediction, in which people may have different operation intentions for the same object on different occasions. Based on the above models, our implicit interaction technology can use the laser pointer to achieve “share attention” with the WMRA and reason the user’s intention. In the end, to execute the household tasks and verify the practicability of the proposed method, we embed the DMP algorithm into the laser interaction technology to generalize the action trajectories of the tasks and achieve the logical state transformation of different actions through the finite state machine.

The contributions of this paper are summarized below:The WMRA with laser interaction is designed, and it can identify the laser semantics correctly even if users point to the wrong position due to hand tremors.The model of intention reasoning and execution based on object affordance is proposed so that the WMRA can identify the desired intention faster and perform tasks with less limb involvement from the user.

The rest of this paper is as follows. Section 2 presents related works. Section 3 describes our system and laser semantic identification. Section 4 describes the technology of reasoning intentions and executing tasks. Section 5 reports the experiments and results. Lastly, the paper is concluded and discussed in Section 6.

## 2. Related Works

### 2.1. Interaction Using Laser Pointer

Interaction using a laser pointer has been widely studied because of its simple operation and clear instruction. Gualtieri et al. created a grasping assistance system for activities of daily living (ADLs), and its novel user interface and grasping capabilities enable this system to grasp objects automatically when a user points to an object with a laser pointer [18]. Sprute et al. employed the virtual boundary to the mobile robot using a laser pointer so that it can navigate according to the user’s wishes [21]. Fukuda et al. used a laser spot to guide the direction of the electric wheelchair and bypassed obstacles successfully [22]. Minato et al. proposed a robot navigation system with the pattern recognition of figures drawn by a laser pointer [23]. Widodo et al. selected the buttons on a large screen by laser spot to control the movement of the robot for people with limited mobility [24]. Kemp et al. proposed a human-computer interaction interface that used a laser spot to navigate the EL_E robot. It allowed humans to select a three-dimensional position in the world and communicate with the mobile robot intuitively, picking up the object selected by the laser spot [25]. Nguyen et al. improved the work of Kemp et al. and realized the designated object’s picking and placement according to the behavior of the laser spot and context environment, in which robots can pick up the designated object from the floor or table, deliver the object to the designated person, and place the object on the designated table [26]. Chavez et al. used the automatic object capture technology to grasp the object locked by the laser spot in an unstructured environment [27].

These studies presented some more user-friendly and low-cost interactive methods with a laser pointer and proved their efficacy. There would not be a heavy burden on the human body when equipped with a laser-pointing device, and users could participate in the process of performing tasks (acquire a sense of engagement). However, robots can only execute some simple tasks when grasping the use of laser interaction in existing research. There is still little discussion about how to further convey complex user intentions and control the robot to complete complex tasks using a laser pointer.

### 2.2. Object Implicit Intention Identification

Object affordance held that people perceived the content in the behavioral possibility provided by things [28]. J. Grezes et al. found that when people perceive the task object, they activate the cortical areas that store visual movement at the same time [29]. From the perspective of neurophysiology, this gave evidence that people perceive objects that can automatically activate action nerves related to objects. The psychological experiments of Anna et al. show that the cognitive system serves for action, and different actions will activate different object constructions [30]. Hassanein et al. believed that the research on object affordance would help to predict the behavior of robots or people, understand social scenes, understand the hidden value of objects, fine-grained scene understanding, and recognize intention [20]. Martijn et al. learned from the concept of object affordance and realized the recognition of the current assembly action and the prediction of the next assembly action according to the object sequence operated by the operator in the video [31]. Mi et al. combined speech recognition with the concept of object affordance and proposed a multi-modal fusion framework based on affordance, which infers the user’s grasping intention according to the user’s voice commands and finally achieves the desired goal [32]. Mo et al. studied the affordance relationship between objects and used the implicit attributes of objects to predict the execution modes of four household tasks [33]. Mandikal et al. embedded the concept of object affordance into the deep reinforcement learning loop to learn grasping policies that are favored by people [34]. Deng et al. made a dataset composed of 18 executable actions and 23 types of objects, which can help the robot identify the implied grasping actions of the objects [35]. Xu et al. studied the expression method of affordance, analyzed the long-term execution effect of objects in the task, and predicted the actions that should be performed in the next step [36]. In addition, some studies have also integrated human actions with object category clues to predict users’ intentions [37,38].

The above research shows that there is a relationship between the object to be operated and the action to be performed. Veronica et al. believed that the research on the intention recognition method based on the idea of object affordance is a new direction to plan more complex tasks using people’s attention and object recognition [39]. However, the revelation relationship between objects and actions in human society is a psychological tendency, not a strict mapping rule. Therefore, the digital modeling of affordance relationships and the adaptability of the model to human habits still hinder the development of implicit interaction.

Kim et al. presented that the robot could perceive the operator’s attention by understanding the user’s gestures, and the robot would identify the objects concerned by the user and infer the task intentions [40]. Kester et al. first proposed an action intention reasoning method based on the category of objects, but the method can only infer actions for a single object without considering the possible impact of the sequence of the concerned objects on action intention recognition [41]. Li et al. observed the user’s eyes through visual technology, inferred the position of the gazed object on the screen, and reasoned the possible subsequent tasks by using the simple Bayes probability model [42]. Wang et al. presented an off-line training and action intention recognition method based on long short-term memory networks that could track eye movement signals such as staring at objects, line of sight inclination, and inclination change rate, which can recognize four sub-action intentions including reach, move, set down, and manipulate [39]. Hitherto, scholars have imitated human’s ability to infer action intention based on the “shared attention” and concept of affordance, which provides a way forward for disabled users to realize implicit intelligent interaction with assistive robots such as the WMRA. However, the existing task reasoning methods based on the concept of affordance are difficult to generalize, and the parsing and execution of tasks mostly depend on expert coding [43]. Therefore, the affordance modeling between objects and actions, the learning mechanism about user preferences and the generalized task execution mechanism need to be further studied.

## 3. Implicit Interaction Using a Laser Pointer

### 3.1. WMRA Specifications

The method proposed in this paper will be developed on the WMRA with an embedded board, NVIDIA Jetson TX2. The WMRA mainly consists of a robotic arm, an electric wheelchair, and other components, and the whole robot is powered by two DC 12 V batteries in series, as shown in Figure 1. The robot is equipped with a Realsense D435i RGB-D camera and a Realsense D435 RGB-D camera. One is a global vision (D435i, eye-to-hand) mounted on the top of the wheelchair, and the other is an operational vision (D435, eye-on-hand) on the robotic arm. It is worth mentioning that D435i, as a global vision, has one more inertial motion unit (IMU) compared to D435, which can measure triaxial attitude Angle and acceleration for the WMRA.

### 3.2. System Overview

In our system, a user could activate the interaction by pointing to a target object using a laser pointer, as shown in Figure 2. The visual system would capture the laser spot, focus on the target object, and extract scene information. The information would be processed by six functional modules:Object recognition: Affordance information of target objects will be accessed, such as visual appearance and spatial relationships.Point cloud data processing: 3D point clouds in the scene are filtered to acquire the objects’ coordinates utilizing the point cloud library.Semantics identification: Identifying the user instructions by the flicker frequency of the laser spot.Intention reasoning: Inferring the user’s intention based on affordance information and semantics information.Grasping posture: An appropriate grasping posture about task intention for the robot arm is generated.Sub-actions reordering: A task expressed by intention consists of some objects and sub-actions. This module will choose and reorder the sub-actions within an action template library and generalize their trajectory.

The touching screen interface enables users to acquire interactive information, such as scene information, user’s intention, confirm and emergency buttons, and operaton of the system. The WMRA will perform the task after the user confirms the robot’s display is the same as their idea.

### 3.3. Semantic Recognition of Laser Pointer Interaction

Precise detection of the laser spot is crucial for the WMRA to focus on the target objects and subsequently identify their semantics. Instead of using conventional background difference or template matching methods [44], we propose a combined algorithm of YOLOv4 [45] and support vector machine (SVM) to detect laser spots and recognize laser semantics. As shown in Figure 3, the visual system would capture the laser spot and transmit the image stream to the YOLOv4 algorithm. Then, SVM gets the input labels and identifies interactive semantics. Finally, the module will output the semantic information that the user chooses a cup and a bowl.

In the process of laser interaction, laser spot flashes regularly on the target objects, as shown in Figure 2c. To calculate the flashing duration and flashing position of the laser spot, the YOLOv4 algorithm, which has fast and stable image recognition capabilities, is used to process the image stream transmitted from the vision system, as shown in Figure 3. In order to enhance the precision of laser spot detection by YOLOv4, an additional 1800 images of the laser spot in a diverse home environment are added to the COCO dataset, which is augmented with random brightness augmentation, random rotation, and salt-and-pepper noise. Considering the potential misreading caused by laser spot shaking due to the limited limb movement abilities of users, we add missed operation samples to the dataset to improve detection accuracy and robustness.

Object category and laser semantic mapping label will be output from the YOLOv4 algorithm and processed to generate a 26-dimensional array, which is defined as the input of the SVM. Four semantic images and labels are respectively presented in the first and fourth rows of Figure 3. If the laser spot is detected but not in the frame of any object, the label array adds 1. If the laser spot is detected in the frame of an object, a numerical label (2 or 3 or 4, etc.) is added based on the serial number of objects. If no laser spot is detected, the label array adds 0.

SVM is known to learn small samples due to the principle of maximum margin [46,47]. To train an SVM classifier for semantic identification, we marked 100 groups of the training set $T=\left\{\left(x_{1},y_{1}),\;(x_{2},y_{2}),\ldots \;,(x_{m},y_{m}\right)\right\}$, in which the input is a 26-dimensional array $x_{i}$ and the output $y_{i}$ is semantic information. For ease of operation, we design three types of semantic information: (1) a short click indicating the selection of an object (numerical label determines the target object) or the user’s decision “yes”, (2) a long press indicates that the object selection is ending or the user’s decision “no”, and (3) a double click indicates reset semantic or end the task in progress. To classify the three types of semantic information, the Gaussian kernel is used to construct dataset features.
(1)$$K\left(x,x_{i}\right)=\exp(-\gamma {\left\|x-x_{i}\right\|}^{2}),$$
where x is the sample value, $x_{i}$ is the landmark of the Gaussian kernel. γ is bandwidth.

After solving the weight parameters of SVM, the complete semantic identification module would recognize the interactive information from a laser pointer.

## 4. Intention Reasoning and Task Execution

The semantic identification module transmits the category information of selected objects to the intention reasoning module, as shown in Figure 2d. To infer the user’s intention, we explore the implicit action of objects and construct an object–action intention network (OAIN) based on the concept of object affordance. Figure 4 illustrates the workflow of the reasoning module. First, OAIN gets semantic information and outputs some intentions with a different probability. Users next express their decision, which will be used to learn user preferences. When the user confirms the intention, the final intention will be output to the next module and logically assembled to execute.

### 4.1. Implicit Action Intention Recognition

The international classification of functioning, disability, and health (ICF) guideline describes the necessary tasks to maintain the lives of people in a home environment. Therefore, we construct a knowledge base dataset about household tasks by extracting the executed object and action labels from the ICF, as shown in Table 1. The dataset contains three types of task descriptions, which consist of a different number of object–action pairs. Objects represent the target that the WMRA will execute, and actions reflect the function affordance of adjacent objects.

In order to formalize affordance between objects, we propose an intention reasoning method based on the conditional random field (CRF). The CRF is a machine learning algorithm that can predict implicit random variables A based on the visible random variable O by solving the maximum conditional probability $P\left(A\left|O\right.\right)$. The way of reasoning action intention with object category information as a context clue can be regarded as a process of solving the maximum probability implicit variable “action intention” with the visible variable “object”.

The CRF algorithm consists of nodes and edges. As depicted in Figure 5, nodes represent the variable, and edges represent the transition features. The edge between object nodes and action nodes represents the state transition feature, which is defined as $s\left(a_{i},o,i\right)$, and the edge between the action nodes represents the action transition feature, which is defined as $t\left(a_{i-1},a_{i},o,i\right)$.

The object and action sets are represented as $O=\left(O_{1},\;O_{2},\;\ldots ,\;O_{n}\right)$ and $A=\left(A_{1},\;A_{2},\;\ldots ,\;A_{n}\right)$, respectively. Under the condition that the value of O is o and the value of A is a. The conditional probability model $P\left(a\left|o\right.\right)$ is expressed as:(2)$$P\left(a\left|o\right.\right)=\frac{1}{Z\left(o\right)}exp\left(\sum_{i,\;k}\lambda_{k}t_{k}\left(a_{i-1},\;a_{i},\;o,\;i\right)+\sum_{i,\;l}\mu_{l}s_{l}\left(a_{i},\;o,\;i\right)\right),$$
where
(3)$$Z\left(o\right)=\sum_{a}exp\left(\sum_{i,\;k}\lambda_{k}t_{k}\left(a_{i-1},\;a_{i},\;o,\;i\right)+\sum_{i,\;l}\mu_{l}s_{l}\left(a_{i},\;o,\;i\right)\right),$$
where $\lambda_{k}$ and $\mu_{l}$ represent the weights of the action transition feature and state transition feature in the model. k and l represent the sequence number of the different transition features. $Z\left(o\right)$ represents the normalization coefficient, which is the weighted sum of all actions and the given objects.

To solve the weight parameter, we use the log-likelihood equation and the improved iterative scaling method. The log-likelihood equation is expressed as:(4)$$L\left(w\right)=log\left(\prod_{o,a}P_{w}{\left(a|o\right)}^{\tilde{P}\left(o,a\right)}\right),$$
where
(5)$$\tilde{P}\left(o,a\right)=\frac{v\left(o,a\right)}{N},$$
where
(6)$$w_{k}=\left\{\begin{array}{c} \lambda_{k},\;\;\;\;\;\;\;\;\;\;\;\;\;\;\;\;\;\;\;\;\;\;\;\;\;k=1,\;2,\;\ldots ,\;K_{1}\\ \mu_{l},\;\;\;\;\;k=K_{1}+l;\;l=1,\;2,\;\ldots ,\;K_{2} \end{array}\right.\;,$$
where $\tilde{P}\left(o,a\right)$ represents the empirical probability distribution. $v\left(o,a\right)$ represents the frequency of a sample $\left(o,a\right)$ in the training dataset. N is the total number of samples.

The prediction of OAIN is the Viterbi algorithm, which can quickly traverse all probability distributions based on the known input sequence and select the maximum probability. The predicted action with the maximum probability can be expressed as:(7)$$a^{*}=arg\underset{a}{\max}w^{T}P_{w}\left(a\left|o\right.\right),$$
where $a^{*}$ represents the optimal action intention sequence.

### 4.2. Learning Mechanism

Although we have trained the OAIN using the CRF algorithm, the OAIN may not correctly output the user’s intention when the user’s preference changes. As shown in Figure 4, in order to adapt to the user preference, we embed the Q-learning algorithm into the intention reasoning module so that the OAIN can automatically update weight parameters after the user makes a decision about action intention.

Q-learning algorithm is a value iteration algorithm based on the Markov decision process in the field of reinforcement learning. The core of Q-learning is making decisions by evaluating the change in Q value. Q value refers to the reward or penalty value after the agent selects the action, and the action triggers a state change for the agent. The Q value will be iterated based on the Bellman equation and stored in the Q table, which consists of state and action.

The low-probability intention reasoning results in the traversal process of the Viterbi algorithm being put into the data cache pool so that the user expresses a decision about the intention through laser interaction, as depicted in Figure 4. After the user locks an intention, the learning mechanism will quantify the interaction process and transmit quantization results to the reward mechanism, which is the Bellman equation in Q-learning:(8)$$Q^{*}\left(s,\;a\right)=\sum_{s\prime }T\left(s,\;a,\;s^{\prime }\right)\left(R\left(s,\;a,\;s^{\prime }\right)+\gamma \underset{a\prime }{\max}Q^{*}\left(s\prime ,\;a\prime \right)\right),$$
where s is the previous state (previous intention), a is action (the process of interaction), $s^{\prime }$ is the current state (current intention), $T\left(s,\;a,\;s^{\prime }\right)$ is a transition function, $R\left(s,\;a,\;s^{\prime }\right)$ is the reward function, and γ is the discount coefficient.

In our scene, the intention of the user may change over time, which means optimal strategy will not exist. Therefore, we adjust the Bellman equation and embed it into our framework. The iterated equation is expressed as:(9)$$q\left(s,\;a\right)=r\left(s,\;a\right)+\gamma q\prime \left(s,\;a\right),$$
where $q\prime \left(s,\;a\right)$ represents the Q value of the previous state; $r\left(s,\;a\right)$ is reward function, the value is equal to 0 and 1 when the user chooses “no” and “yes”. respectively; γ is discount coefficient and set as 1.

### 4.3. Intention Execution

To perform actions intention sequence continuously, the finite state machine (FSM) is used to logically assemble actions. The FSM is a state transition mechanism, which has the characteristics of state storage and logic coordination. Figure 6 illustrates an example of a “grasp cup pour bowl” task. Firstly, the WMRA will process the point cloud data of the target objects to calculate their positions. The WMRA will choose the template trajectory based on the action intention from our action template library and generate new trajectories to adapt to the positions of target objects in the scene. Lastly, trajectories would be reordered by the FSM, and the WMRA follows FSM-defined rules to execute action intention automatically.

Given the demonstrated trajectories in the template library cannot adapt to the change of object position in the scene, the dynamic movement primitives (DMP) algorithm [48,49] is used to post-process the trajectories. DMP is a known trajectory generalization algorithm [50], which would further generalize the trajectory of similar actions by solving the optimal curve characteristic solution of the teaching trajectory. The generalization of one-dimensional motion is generated by the following differential equations.
(10)$$\tau \dot{v}=K\left(g-x\right)-Dv+K\left(g-x_{0}\right)s+Kf\left(s\right),$$
where
(11)$$\tau \dot{x}=v,$$
where x is the position of the trajectory. v is the speed of the trajectory points. $x_{0}$ is the starting point of the trajectory. g is the goal of the trajectory. τ is the time scaling factor. K and D are the gain coefficients of the system, which are used to adjust the convergence trajectory of the system. $f\left(s\right)$ is the nonlinear function of the demonstrated trajectory.

## 5. Experiment and Results

In this section, the feasibility of the proposed implicit interaction technology is verified on a WMRA platform. We test the ability of the laser interaction, the intention reasoning model, the learning mechanism, and task execution. Compared with some previous works [41,44], our model achieves some better results.

### 5.1. Laser Interaction Evaluation

We have deployed YOLOv4 and SVM-based laser interaction on the embedded board, Jetson TX2. Compared to other visual algorithms, such as Transformer [51], the YOLOv4 algorithm requires fewer computational resources, has faster recognition speed, and is more compatible with hardware environments. Therefore, the YOLOv4 is selected as our visual detection algorithm.

As shown in Figure 7, researchers conduct 50 clicking tests on different objects respectively and observe whether the output results match the target objects. The success rate is shown in Table 2. The results represent that the minimum success rate can reach over 92%, with the accuracy of the table even reaching 100%. We analyze that the high success rate is due to their larger clickable space, more uniform surface curvature, and less laser reflection, while the objects with lower success rate do not possess these characteristics.

Then, we test the accuracy of SVM for semantic recognition and construct a neural network (NN) for comparison, which includes three fully connected layers, and the activation function is Rectified Linear Unit (ReLU). Before the test, we created a guide that describes some target semantics to be implemented (Each semantic is repeated 20 times in a random order in the guide). Researchers click on the laser pointer to output the semantics and judge whether the result is consistent with the content of the guide. The recognition accuracy is shown in Table 3.

A timing-based method is used as a baseline, which recognizes the laser semantics by calculating the time the laser spot appears. The results show that the SVM and NN methods are much better than the baseline. However, compared to the black-box feature of NN, SVM has a rigorous mathematical theory and is a more transparent classification algorithm. Therefore, we combine the YOLOv4 and SVM algorithms to recognize the laser semantics.

### 5.2. Intention Reasoning Evaluation

In order to compare the ability of intention reasoning with the previous work [41], which combines the Markov network and Bayesian incremental learning, we select the same objects and actions to reasoning intention as shown in Table 4.

Under the same conditions of quantity and type of objects and actions, our work can output 36 single-object intentions and 19 multi-objects intentions. In the other work, the reasoning model could only output the intention with an object and an action, which is low relative to our model. Compared to their work, we can output intentions for several objects and actions. In our opinion, more intentions reflect that the robots have a better comprehension of users and afford users an increased range of options.

### 5.3. Learning Ability Evaluation

The main metric for evaluating performance is the number of human-computer interactions in each session. The object in the scene selected by the user for the first time is defined as entering the first session, and the laser semantic “yes” or “no” of the user output decision is defined as human–computer interaction. In the process, the user may judge the target intention and find the desired intention in the first session through multiple human-computer interactions. When the user selects the same objects for the second time, it is defined as the second session, and the user may only need less human-computer interaction time to determine the target intention in the second session. The ideal situation will appear when the number of interactions is one, and it is indicating that the first query to users is their expected intention. Table 5 further lists the various groupings of intentions that are tested in the evaluation.

As shown in Figure 8, it requires four interactions when our method outputs the user’s expected intention for the first time. In the span of twenty sessions, the average number of interactions required by our framework decreases monotonically. Finally, the average value of all groups will be equal to one. After twenty sessions, the average reduction in interaction is 70%. Because we set the same scene and intention as the previous work and the number of sessions to the desired intention is a 75% reduction, we believe that our method could learn user preferences and be superior.

Besides, we also evaluate our method through the intention composed of multiple objects, and to our knowledge, the previous work has not achieved this. As shown in Table 6, we set two evaluation groups in which task intentions are composed of two objects and three objects, respectively. The metric and process of evaluation are similar to the above.

As shown in Figure 9, the desired intention in the first session required more interactions compared with the single object intentions. Moreover, it can be seen that the desired intention of three objects required more time interaction than the two objects in the first session. We analyze that the intention prediction of the Viterbi algorithm is to calculate the probability layer by layer, and each layer has many invalid intentions. However, by constantly learning the right intentions, the number of interactions reduces and reaches the ideal situation. Therefore, in general, our method can identify the user’s intention and adapt to user preferences successfully.

Furthermore, we record the time spent on intention reasoning. As we can see from Table 7, the average intention recognition time of a single object is about 0.12 s, and the average intention recognition time of multiple objects is about 0.19 s, which is less than 0.3 s and meets the requirements of real-time recognition.

### 5.4. Household Task Execution Evaluation

To evaluate the execution ability of laser implicit intention interaction technology on the WMRA platform, we record and analyze the process of performing tasks, and compare our method with joystick operation.

As illustrated in Figure 10, we take the task “grasp cup pour bowl” as an example and calculate the proportion of human–computer interaction in the video stream. The interaction time required by our method is about 17 s, accounting for 14.78% of the total time. However, in manual mode, the user needs to continuously control the joystick when performing household tasks. Although we only list the interaction time required for the task “grasp cup pour bowl”, the time required for other tasks will not change greatly unless the amount of objects change. Therefore, compared with the joystick operation, the proposed technology could not only save time for task execution but also significantly reduce the user’s limb involvement time (about 85%), which effectively alleviates the user’s physical burden.

## 6. Conclusions and Discussion

To improve the control sense, usability, and adaptability of the WMRA, we propose an object affordance-based implicit interaction technology using a laser pointer. A laser pointer could lock the target objects, similar to the human gaze. Therefore, a YOLOv4 and SVM-based laser semantic identification algorithm is proposed to convey user intention intuitively. Moreover, the robustness of the algorithm contributes to the automatic correction of the wrong semantics when the user points to a wrong position because of hand tremors. Next, we were inspired by the cognitive psychology of human interaction processes and designed an object affordance-based intention reasoning algorithm, which is based on CRF and Q-learning. The WMRA realized the implicit action intention identification of target objects and learned user preference through the decision of user interaction. Lastly, in order to execute the action sequence in the scene, trajectories of actions were generalized by the DMP and transmitted to the FSM to logically reorder.

We selected many objects and conducted the click test of laser interaction. The results show that the lowest success rate of correctly selecting a desired object is 92%, and the highest is 99%. We inferred that the impacts on the success rate are the surface curvature and material, which is consistent with the discussion in [24]. However, compared to previous works on laser interaction [22,23,24], we designed more forms of laser clicking and achieved more laser semantics through intention recognition.

Based on the experiments and results, we believe that our research could reduce the burden of interaction between disabled users and assistive robots. Users only need to sequentially click on the target objects, and then they can wait for the robot to execute automatically. Due to security considerations, the research did not include human-related tests. However, in our opinion, the less participation during the task execution, the fewer the users’ interaction burden. Overall, this paper realizes the conveyance of complex task intention through simple interactive operation and executes it, which lays a technical foundation for the rapid practical application of the WMRA.

## 7. Limitations and Future Work

This paper explores an implicit interaction system that can assist users in completing household self-care tasks, which can reduce the interaction burden in the operation of the robot. However, some limitations that affect our research should be noted. For safety reasons, we have not conducted tests on older adults or other individuals, which limits the diversity of experimental samples. To achieve interactive experiments between robots and older adults, the reliability of the proposed system and the psychological pressure that robots bring to users, such as the trusted testing between users and robots [52], need further research.

Based on the above limitations, we should not only focus on the execution ability of the robots but also consider the cognitive and decision-making burden of users during the robot operation process. These provide ideas for future research. As reviewed by [53], Ergonomics & Human Factors (E&HF) should be taken into account in the design process of robots. E&HF contributes to understanding users’ capabilities and limitations and uses this information to minimize the cognitive burden on humans in the process of operating robots [54]. Therefore, we will further study the reliability and ergonomics of our laser interaction systems and test their ability to interact with older adults in future work.

## Figures and Tables

**Figure 1 sensors-23-04477-f001:**
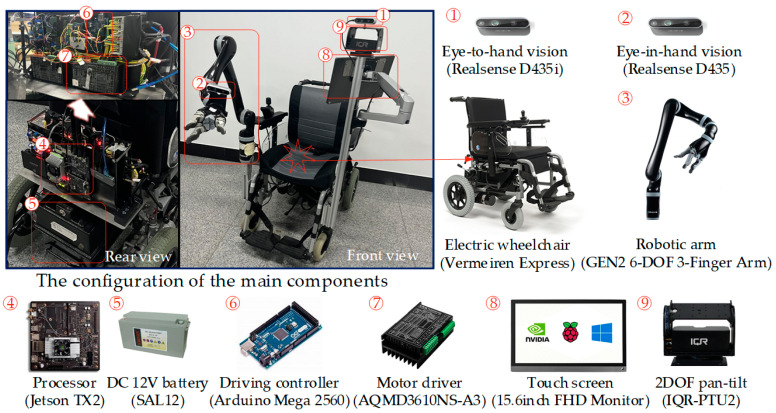
Specifications of the WMRA.

**Figure 2 sensors-23-04477-f002:**
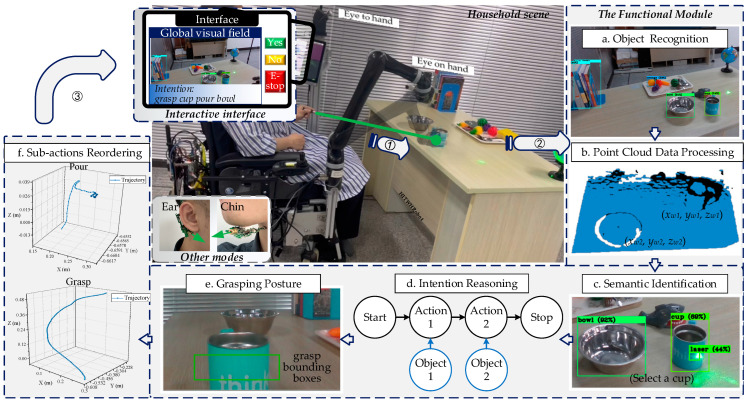
The program flow chart of the interaction system with a laser pointer (green laser).

**Figure 3 sensors-23-04477-f003:**
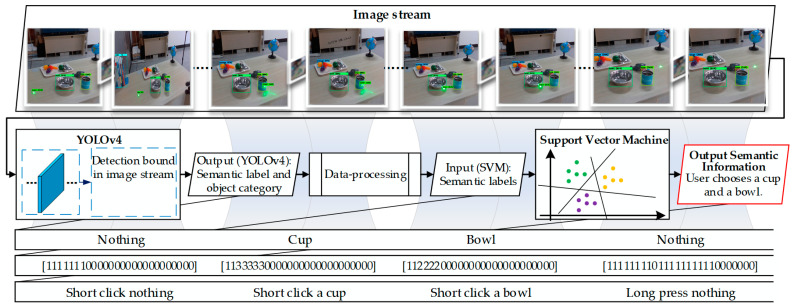
The principles diagram of semantic identification of laser interaction.

**Figure 4 sensors-23-04477-f004:**
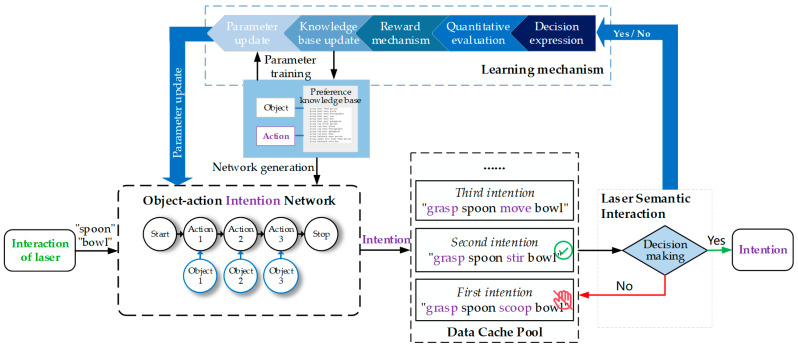
The schematic of intention reasoning with a preference learning mechanism.

**Figure 5 sensors-23-04477-f005:**
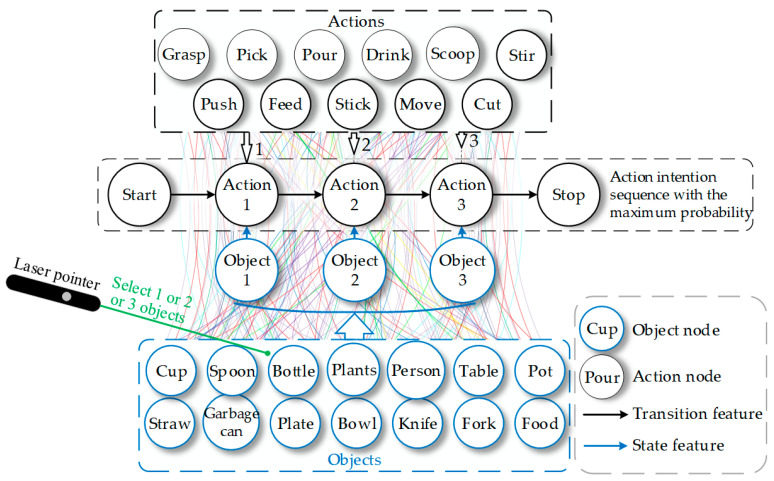
Object–action intention network model. Blue and black circles represent objects and actions in the dataset, respectively. Colored lines represent probabilistic relationships between objects and actions. Object 1, object 2, and object 3 are selected targets by a user with a laser pointer. Action 1, action 2, and action 3 are the output action intention sequences based on the objects.

**Figure 6 sensors-23-04477-f006:**
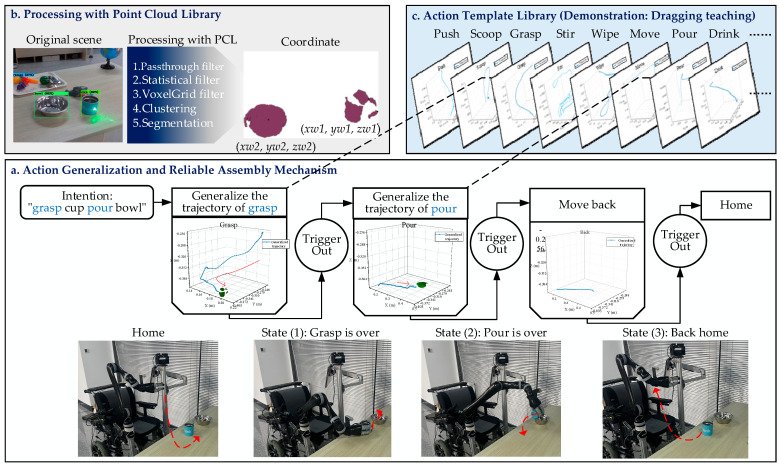
Intention execution mechanism. (**a**) The workflow diagram of the FSM. (**b**) The point cloud data of the selected targets are processed, and the filter functions in PCL are used to calculate the position coordinates of the targets in the scene. (**c**) A library of action templates that we build by dragging teaching.

**Figure 7 sensors-23-04477-f007:**
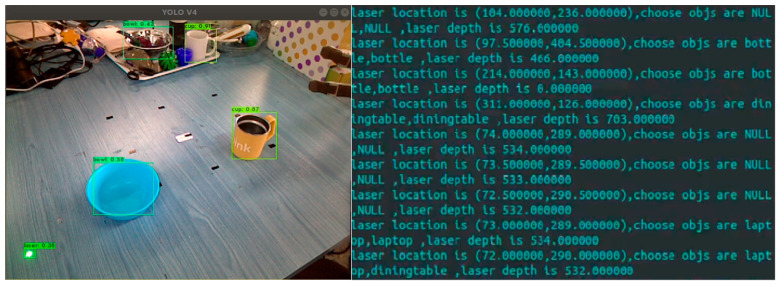
Example of the experimental procedure for clicking test. The left picture shows that the researcher points a laser spot on the table. The right picture shows that our system is recognizing the laser spot position and selected objects.

**Figure 8 sensors-23-04477-f008:**
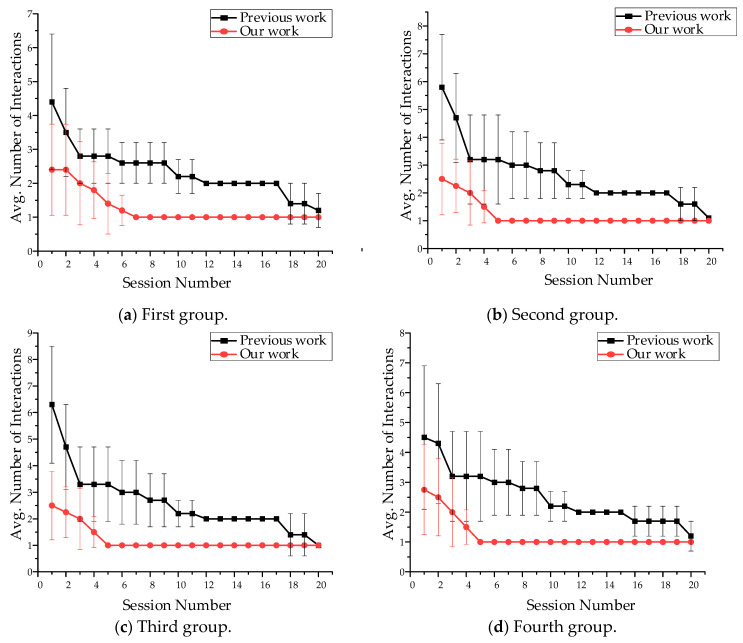
The result of the single-object intention interaction. (**a**–**d**) The node represents the average number of interactions in each group, and the line represents the standard deviation in the number of interactions required by each intention in the group.

**Figure 9 sensors-23-04477-f009:**
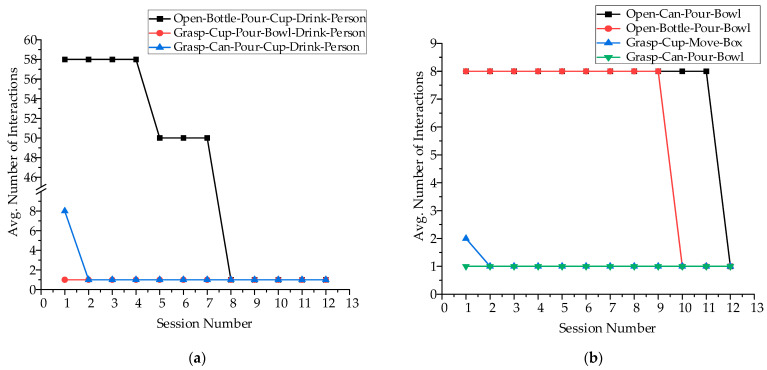
The result of the multi-objects intention interaction. (**a**) First; (**b**) Second.

**Figure 10 sensors-23-04477-f010:**
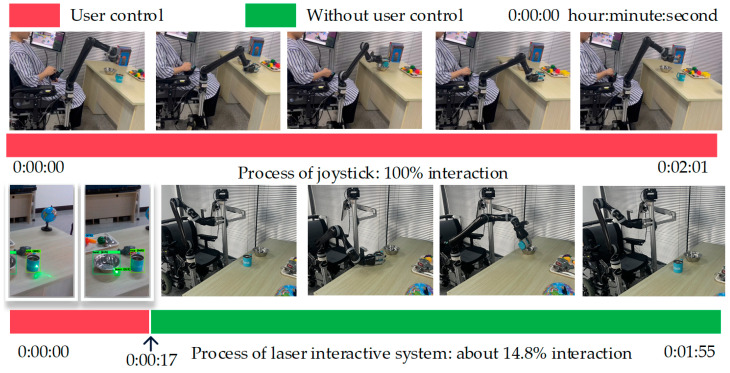
Comparison diagram of limb involvement during the handle interaction and laser interaction.

**Table 1 sensors-23-04477-t001:** Partial examples from our knowledge base dataset ^1^.

Single Object (a_1_, o_1_)	Two Objects (a_1_, o_1_, a_2_, o_2_)	Three Objects (a_1_, o_1_, a_2_, o_2_, a_3_, o_3_)
pick cup	grasp cup pour plants	pick spoon scoop bowl feed person
grasp cup	grasp cup pour bottle	pick spoon scoop plate feed person
push cup	grasp cup drink person	pick spoon scoop pot feed person
……	……	……

^1^ For more detailed information, please contact the corresponding author.

**Table 2 sensors-23-04477-t002:** The result of the clicking test.

Object	Success	Fail	Success Rate
Table	50	0	100%
Cup	49	1	98%
Bottle	49	1	98%
Bowl	47	3	94%
spoon	46	4	92%

**Table 3 sensors-23-04477-t003:** The result of the semantics recognition test.

Object	Timing-Based	NN	SVM
Short click	80%	95%	95%
Long click	5%	90%	90%
Double click	65%	90%	95%

**Table 4 sensors-23-04477-t004:** Experimental dataset. Previous work [41] chose 11 objects and 7 actions to evaluate the ability of intention reasoning and preference learning. We, therefore, select the same objects and actions to compare with their work ^1^.

Category	Dataset
Objects (11)	Bottle, Bowl, Box, Can, Carton, Cup, Mug, Spray-can, Tin, Tube, Tub
Actions (7)	Drink, Grasp, Move, Open, Pour, Push, Squeeze
Object-action intention (36)	Drink-Bottle, Grasp-Bottle, Move-Bottle, Open-Bottle, Pour-Bottle, Grasp-Bowl, Move-Bowl, Push-Bowl, Grasp-Box, Move-Box, Open-Box, Push-Box, Drink-Can, ……
Objects-actions intention (19)	Grasp-Carton-Move-Box, Grasp-Can-Move-Box, Grasp-Bottle-Move-Box, Grasp-Box-Move-Box, Grasp-Cup-Move-Box, Grasp-Can-Pour-Bowl, Grasp-Can-Pour-Cup, ……

^1^ For more detailed data, please contact the corresponding author.

**Table 5 sensors-23-04477-t005:** Single-object intention evaluation groups. Single-object intentions consist of an object and an action. To compare with [41], we choose the same four sets of intentions.

Group	Intention
First	Grasp-Box, Open-Box, Grasp-Carton, Open-Carton, Pour-from-Carton
Second	Grasp-Can, Move-Can, Pour-from-Can, Drink-from-Cup
Third	Grasp-Carton, Move-Carton, Pour-from-Carton, Drink-from-Cup
Fourth	Grasp-Bottle, Move-Bottle, Pour-from-Bottle, Drink-from-Cup

**Table 6 sensors-23-04477-t006:** Multi-objects intention evaluation groups.

Group	Intention
First	Grasp-Cup-Move-Box, Grasp-Can-Pour-Bowl, Open-Can-Pour-Bowl, Open-Bottle-Pour-Bowl
Second	Grasp-Can-Pour-Cup-Drink-Person, Grasp-Cup-Pour-Bowl-Drink-Person, Open-Bottle-Pour-Cup-Drink-Person

**Table 7 sensors-23-04477-t007:** Time of objects–actions intentions reasoning.

Single-Object	Time(s)	Multi-Objects	Time (s)
Cup	0.146924	Cup, Bowl	0.210154
Bowl	0.185253	Cup, Box	0.182532
Can	0.180976	Bottle, Box	0.188147
Box	0.143749	Bowl, Cup, Person	0.189620
Bottle	0.165264	Spoon, Bowl, Person	0.166144

## Data Availability

Not applicable.

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
