# Peer review of "Object Affordance-Based Implicit Interaction for Wheelchair-Mounted Robotic Arm Using a Laser Pointer"

_sensors, 2023, doi:10.3390/s23094477_

Round 1

Reviewer 1 Report

The manuscript presents an object affordance-based implicit interaction technology using a laser pointer to support the interaction between older users and a wheelchair-mounted robotic arm. The research topic is novel, interesting and under the Sensors journal's scope.

The reviewer suggests some improvements before publication:

Line 171: The subchapter should not begin with a figure, and the Figure 1 have to be referred in the previous text (before the figure inclusion).

In methodology description, a sample characterization should be included.

Results Discussion: This is the main concern! The authors have to improve the discussion section, comparing their results with previous studies and identifying limitations/future work. For example: 

- the authors did not perform a usability and users' confidence assessment/testing, as performed in this example: https://doi.org/10.3390/robotics11030059

- the design of collaborative robotics, mainly in this scenario, should consider Ergonomic and Safety requirements, the authors could discuss this (some examples of previous studies: doi.org/10.1016/j.rcim.2020.101998  https://doi.org/10.3390/safety7040071 

A section focused on limitations and future research shall be included. 

Reviewer 2 Report

In the introduction and conclusion you focused on the elderly but there is no test or significant evidence with this population. Moreover you "ensure" with the algorithm that the pointing position is corrected but there is no evidence of that.

"If the joystick were to be operated by older adults with limited limb  movement abilities, the task completion time would substantially increase" are statements that lack evidence but only show a conjecture or opinion. This kind of statements need to be avoided. 

There is no enough justification for using YOLOv4 and the SVM classifications. What are they advantages with other methods or a formal comparison should be added. 

About the Q-learning algorithm, only the general description of the algorithm is described but the actual reward functions and evaluation is missing. 

The experiments should include different users and a further analysis. 

In general, is a good project but lacks a scientific methodology that affects how the paper is presented. 

Round 2

Reviewer 1 Report

The authors improved the manuscript, clarifying the topics pointed out by the reviewer. Congrats!

Author Response

Dear Respected Reviewer:

We are deeply grateful for your work. The previous comments have been extremely helpful in improving our manuscript.

Best regards,

Yan Liu

Reviewer 2 Report

I understand you are not allowed to conduct experiments, but mentioning a specific application for older adults is misleading because there is no actual evidence of your system being helpful for them. Please remove that part of older adults or provide sufficient evidence to validate that your system effectively can do what you say it does. 
